# The Preparation of Amorphous ZrC/Nanocrystalline Ni Multilayers and the Resistance to He^+^ Irradiation

**DOI:** 10.3390/ma15093059

**Published:** 2022-04-22

**Authors:** Shengming Jiang, Ruihua Zhu, Xiaotian Hu, Jian Zhang, Zijing Huang

**Affiliations:** College of Energy, Xiamen University, Xiamen 361005, China; 32420210156625@stu.xmu.edu.cn (S.J.); 32420171152092@stu.xmu.edu.cn (R.Z.); 32420201152830@stu.xmu.edu.cn (X.H.)

**Keywords:** accident-tolerant materials, amorphous ZrC, nanocrystalline Ni, resistance to He^+^ irradiation

## Abstract

The development of accident-tolerant materials is of great significance for preventing the zirconium–water reactions and improving the inherent safety of nuclear reactors. In this study, ZrC/Ni multilayers with average layer thicknesses of 5, 10, 20, 50, and 100 nm were designed and successfully fabricated by magnetron sputtering. The characterization results of GIXRD, SEM, AFM, TEM, etc., show that the series of films are mainly composed of alternately deposited Ni crystalline layers and ZrC amorphous layers, and the interface is clear. The films were irradiated with 50 keV He^+^ with a fluence of 1.0 × 10^17^ ions/cm^2^ at room temperature, and the films with different layer thicknesses kept the original phase composition. It was found that an amorphous transition layer with a thickness of about 30 nm appeared between the amorphous and crystalline interface of the 100 nm film by TEM characterization. The analysis shows that this layer is formed by the mixing of Ni and Zr elements induced by irradiation, which is not conducive to He^+^ migration and produces large-sized helium bubbles. The appearance of the transition layer improves the irradiation stability of the amorphous/crystalline composite film, thus providing a theoretical basis for the application of this type of material in fuel cladding.

## 1. Introduction

The accident at the Fukushima nuclear power plant in Japan in 2011 exposed the safety problem of zirconium alloy fuel cladding that easily releases hydrogen due to the zirconium–water reaction under high-temperature water vapor. In order to improve the safety performance of nuclear fuel, a series of “accident tolerant fuel” (ATF) technical concepts have been proposed [1,2,3,4,5]. The surface coating technology stands out for its mature technology and wide application [6,7]. Film performance is a key factor in coating technology. The traditional films have single structures and poor comprehensive performance, making it difficult to adapt to the increasingly complex service environment. Emerging multilayer films, especially nano-multilayer films, have a variety of excellent properties and can be used in harsh environments, gradually attracting the interest of researchers [8,9]. The properties of nano-multilayer films are mainly dependent on deposition conditions, interface composition, and type [10,11]. Common crystalline types include face-centered cubic (FCC), body-centered cubic (BCC), and hexagonal close-packed (HCP). The researchers developed a number of metal multilayer films with homo-crystalline-type interfaces (FCC/FCC [12,13] and BCC/BCC [14]) or hetero-crystalline-type interfaces (FCC/BCC [15,16], FCC/HCP [17], and BCC/FCP [18,19,20]). As far as the radiation resistance is concerned, the existence of the interface increases the energy of the system and acts as a defect sink [21]. Irradiation-generated interstitial atoms or vacancies diffuse to the interface and are prone to pinning, thereby promoting the agglomeration of smaller defects into larger defects. For example, helium grows at the interface to form helium bubbles [22,23], or interstitial atoms migrate to the interface and annihilate directly [24,25]. As a result, the energy of the system is reduced, and this is helpful to the structural stability of the multilayer films.

In addition, ceramics have the advantages of high corrosion resistance and high hardness, which can promote the mechanical properties and irradiation properties of multilayer films [26]. Previous studies have focused more on different types of ceramic/ceramic nano-multilayer films, while a single ceramic system multilayer film is prone to brittle fracture in practical applications. Studies have shown that the introduction of metal layers can significantly improve the toughness, bonding strength, and wear resistance of nano-multilayer films [9]. Recent findings demonstrate that amorphous/crystalline nano-multilayer films exhibit good irradiation stabilities due to the presence of amorphous layers and abundant amorphous/crystalline interfaces [27]. Zirconium carbide (ZrC) has excellent physical and chemical properties that allow it to be used in a variety of extreme conditions, such as high-temperature materials for aerospace applications and fuel cladding coatings in nuclear reactions [28,29,30]. In addition, due to its high hardness and wear resistance, it is often used as a protective coating for cutting tools [31]. Nickel (Ni) has good corrosion resistance and high-temperature oxidation resistance, and the nanocrystals formed in Ni films have excellent density and structural stability, so they are often widely used as protective coatings [32,33].

In this paper, the magnetron sputtering method was used to deposit alternately ZrC amorphous layers and metal Ni crystalline layers on the surface of Si substrates in an Ar atmosphere of 0.7 Pa at room temperature. A series of ZrC/Ni nano-multilayer films with uniform total thickness and different monolayer thicknesses were prepared. He^+^ irradiation experiments were conducted on the prepared films to explore the effect of different layer thicknesses on the structural stability of nano-multilayer films.

## 2. Experimental Procedure

### 2.1. Sample Preparation

Amorphous ZrC/nanocrystalline Ni multilayer films were prepared on single-sided polished Si (111) substrates by magnetron sputtering (Denton Desktop Pro). Ni single-element metal target (99.99%) with direct current (DC) sputtering and ZrC ceramic target (99.95%) with radio frequency (RF) sputtering were used. Before the deposition, the Si substrates, cut into square pieces of 10 mm × 10 mm, were cleaned with acetone and absolute ethanol by ultrasonic vibration for 30 min, and then dried for 24 h at 70 °C. Dried Si substrates, fixed on a disc base 80 mm from the target, were pre-sputtered for 3 min in the chamber of magnetron sputtering instrument to remove surface contaminants. The chamber pressure was first pumped to below 10^−5^ Pa and then maintained at 0.7 Pa by adjusting the flow of high-purity argon (99.999%) at room temperature during the deposition. A Ni layer was first deposited on the Si substrate with a sputtering power of 80 W, at a sputtering rate of 0.91 Å/s, followed by a ZrC layer with a sputtering power of 100 W, at a sputtering rate of 0.18 Å/s. In order to ensure the uniformity of sample growth during magnetron sputtering, the disc base was always rotated at a constant speed of 20 rpm. The films were all deposited in the order of Ni-(ZrC-…-Ni)-ZrC, with the different design thicknesses (T_d_) for each layer, including 5, 10, 20, 50, and 100 nm. Table 1 shows the sputtering parameters.

### 2.2. Ion Irradiation and Characterization

The prepared samples were irradiated with 50 keV He^+^, using a 400 kV ion implanter from National Electrostatics Corporation (NEC) installed at Xiamen University. All films were irradiated with the irradiation flux of ~1.0 × 10^13^ ions/(cm^2^·s) at room temperature. The irradiation fluence was 1.0 × 10^17^ ions/cm^2^. The high fluence irradiation not only allows us to study the resistance of the film under high irradiation levels, but is also helpful to observe the aggregation of helium in the Ni layer at room temperature [34,35]. Prior to this, SRIM calculation was perform to judge the distribution of radiation damage. The threshold energy of Zr is 37 eV, that of C is 16 eV, and that of Ni is 40 eV [36,37]. The results showed that most of the damage was concentrated in the multilayer when 50 keV He was irradiated to 1.0 × 10^17^ ions/cm^2^. The depths of the damage peak and helium concentration peak are about 150 nm and 180 nm, respectively. Especially for a film with single-layer thickness of 100 nm, the radiation damage and helium concentration distribution in the range of 30 nm near the interface are 2.4–2.5 dpa and 2.3–3.1 at%, respectively. Therefore, the damage in this area is considered to be consistent.

The phase composition of the multilayer films was characterized by glancing incidence X-ray diffraction (GIXRD, Rigaku Ultima IV), with a step of 0.02° and an incident angle of 0.5°. The surface morphology of some multilayer films was investigated by atomic force microscopy (AFM, Dimension FastScan) and field emission scanning electron microscopy (FESEM, SUPRA 55). The microstructures of the multilayer films were analyzed by transmission electron microscopy (TEM, Tecnai F30 TWIN), with an accelerating voltage of 300 kV, while the nanobeam electron diffraction (NBED) technique was applied to characterize a circular region with a diameter of ~60 nm for diffraction pattern analysis. Scanning transmission electron microscopy (STEM) and energy-dispersive X-ray spectroscopy (EDS) were also used for sample composition analysis. TEM specimens were prepared by mechanical polishing and ion milling, using a precision ion polishing system (PIPS GATAN PIPS II 695) selected with low energy (5–3 keV) and low angle (4–2°). 

## 3. Results and Discussion

### 3.1. Original Structure Characterization

Figure 1 shows the GIXRD spectra of pristine amorphous ZrC/nanocrystalline Ni multilayers with different single-layer thicknesses, and the Ni grains size in different films was calculated. It is worth noting that, for all nano-multilayer films, no Zr-containing crystal phase was found in Figure 1a, only a bulging diffraction peak near 30° and a unique set of diffraction peaks matched with the Ni PDF card (PDF#62-2865). This shows that the Ni layers deposited by magnetron sputtering are crystalline layers, while the ZrC layers exist in amorphous state, which also indicates that the phase composition of each layer of nano-multilayer films prepared by magnetron sputtering is uniform and consistent. Here, the grain size of the crystalline Ni layers is calculated by using the Scherer formula [38]:(1)$$D_{\mathrm{hkl}}\;=\;\frac{K\lambda }{\beta \cos\theta_{\mathrm{hkl}}}$$
where $D_{\mathrm{hkl}}$ is the dimension of the crystal in the (hkl) orientation; K is the half-width Scherrer constant, taking 0.9 [20], since the Ni grains are irregular polyhedrons; λ is the X-ray wavelength; β is the full width at half maximum (FWHM); and $\theta_{\mathrm{hkl}}$ is the Bragg reflection angle in (hkl) orientation. The calculated crystalline sizes for the (111) plane orientation are plotted in Figure 1b. Actually, the FWHM of the X-ray diffraction peaks on different crystalline planes contain the shape characteristics of the crystalline grains in this orientation [39]. The smaller the FWHM, the larger the grains sizes for a certain crystalline material, and this reflects the growth of grains in a preferred orientation [40,41]. To further illustrate the relationship between the shape characteristics of the Ni layer grains and the layer thickness, D_(hkl)_ was calculated and summarized in Table 2. From the calculation results, the growth rates of Ni grains on different crystal planes are different. As shown in Table 2, D_(200)_ and D_(220)_ are less than D_(111)_. This result indicates that the grain growth of the Ni layers has a preferential (111) orientation during the magnetron sputtering process. As the most densely packed plane in the FCC structure metal is the (111) plane, when the metal film is formed, the (111) plane will be parallel to the film surface with the minimum surface energy [42]. Therefore, it can be judged that the normal line of the (111) plane is parallel to the direction of the film thickness. The XRD results also show that the normal direction of the (111) plane is the best orientation for the Ni grain. In this paper, the thickness along the normal direction of the (111) plane is used as the size of the crystal grain to discuss the influence of irradiation on grains growth. It is easy to find that crystalline size, D, in the size range of 2.97–9.6 nm, gradually increases with the increase of monolayer thickness, T_d_, in Figure 1b.

Based on the theory of evolutionary overgrowth during film thickening [43], there is a non-linear relationship between grain size (D) and layer thickness (Tr), D = A × Tr^n^ (growth exponent *n* and pre-factor A) [44]. We performed non-linear fitting on the crystal-size data to obtain the fitting formula D = 1.59 × Tr^0.39^ in Figure 1b. The way that films’ grains grow depends primarily on the homologous temperature, which is the ratio of the substrate temperature to the melting temperature of the material. The crystals grow mainly by the way of grain boundary migration with the growth exponent *n* = 0.35 ± 0.04 when the homologous temperature is between 1.7 and 2.6 [45]. In this experiment, the substrate temperature was kept at room temperature during the deposition of the Ni metal layer; the homologous temperature was ~0.17, and the growth exponent, *n*, was ~0.38, which is basically consistent with the literature data. The growth of Ni crystals obeys the principle of minimum interface and surface energy, and the small crystal grains gradually shrink or disappear through the movement of the interface. This growth model will result in grain size growth as the film thickness increases at a constant sputtering rate [42,46,47,48].

Figure 2 shows cross-sectional SEM images of pristine amorphous ZrC/nanocrystalline Ni multilayer films. Delamination of films with a monolayer thickness of 50 nm is demonstrated by bilayer films. Regardless of the layer thickness of the multilayer films, the interfaces between the layers are clearly distinguishable, and each layer is sufficiently straight in this field of view. This reflects that the multilayer films formed by this process are of good quality, uniform in composition, and distinct in structure, thus providing a basis for subsequent research on helium ion irradiation. Based on the SEM image, the actual average layer thickness of each multilayer films was measured as shown in Table 2. Comparing the preliminary design and the actual layer thickness, it is found that the layer thickness obtained by magnetron sputtering maintains a small enough difference with the expected value. Films with layer thicknesses of 10 and 100 nm were selected for AFM characterization (shown in Figure 3), from which the surface roughness, Rq, of the films was obtained as 1.476 and 1.046 nm, respectively. This still demonstrates that the prepared films are of good quality. Therefore, it can be basically considered that the ideal films are obtained experimentally.

In an ideal state, the membrane materials used for the fuel cladding in a nuclear reactor, as the accident-tolerant materials, need to have high enough irradiation resistance to avoid the harm caused by the zirconium–water reaction. Therefore, it is necessary to perform ion irradiation tests on the films to evaluate their ability to resist irradiation damage. Prior to this, a multilayer film with a single-layer thickness of 100 nm was selected as the main research object for preliminary characterization, as shown in Figure 4. Figure 4a shows the TEM bright-field mode (BF) image of the cross-sectional sample and the NBED patterns of the corresponding layer. The contrast in the bright-field image reflects the approximate morphology of the crystals. It can be seen that the nanocrystals of the crystalline Ni layer are basically arranged in stripes. The diffraction pattern of the crystal layer is composed of rings with different radii formed by many bright spots, which confirmed that the crystal layer is composed of Ni nanocrystals. The diffraction pattern of the amorphous layer corresponds to a diffuse halo. These results are consistent with those obtained from the previous XRD data. Figure 4b,c shows the TEM bright-field image and HRTEM image of the area near the interface, respectively. There is a significant difference in the contrast between the two layers, but the interface width cannot be identified under this multiple. The HRTEM image also shows reduced contrast at the interface. The interface is more likely formed by the densely filling of a short-range ordered amorphous structure and crystals with preferential growth orientation. This corroborates the element mixing phenomenon at the interface. In summary, the film with a single-layer thickness of 100 nm has an approximately ideal composition and structure, and we can continue to study the microscopic evolution mechanism of the film under He^+^ irradiation.

### 3.2. He^+^ Irradiation Characterization

Figure 5a is the GIXRD pattern of five multilayer films irradiated with He^+^ at room temperature. Compared with Figure 1a, it was found that the corresponding peaks of all ZrC/Ni multilayer films remained unchanged even under high-fluence He^+^ irradiation of 1.0 × 10^17^ ions/cm^2^, indicating that the samples still maintain the structure of crystalline Ni layers and amorphous ZrC layers. Figure 5b is the calculated Ni crystal grains size as a function of monolayer thickness based on the GIXRD data after irradiation. The grain size of the films after irradiation was between 4.08 and 11.35 nm, which is larger than that for pristine samples, thus revealing that He^+^ irradiation promoted the recrystallization growth of Ni nanocrystals. This phenomenon can be attributed to the annealing effect of He^+^ irradiation. When the energy-carrying He^+^ is projected into the samples, most of the energy is transferred to the atoms around the ion track in the form of electron energy loss, causing the originally dispersed atoms to undergo energy transitions and tend toward forming a lower-energy crystal structure [49,50]. 

The film with a layer thickness of 118.9 nm was further characterized by transmission electron microscopy. Figure 6a is a BF image of the overall morphology at low magnification. It is obvious that the film remains intact after 1 × 10^17^ He^+^/cm^2^ irradiation, meaning that the film can withstand the high fluence of He^+^ irradiation and maintain the stability of the overall structure. In actual working conditions, the well-structured film can effectively isolate the contact between Zr and H_2_O, thus fundamentally eliminating the harm caused by hydrogen explosion. Careful observation reveals that there is a contrast difference between the crystalline Ni layer and the amorphous ZrC layer at the interface. By further increasing the magnification, it can be found that a new layer with a thickness of about 30 nm appears at the interface of the irradiated film, named the transition layer. The diffraction patterns corresponding to the amorphous and crystalline layer are a diffusion halo and bright spots, respectively, which are unchanged compared to Figure 6b. It shows that the irradiation does not cause phase transition, thus reflecting the great ability of the film to resist He^+^ irradiation. Figure 6d shows the location of the selected region of the EDS surface scan, and Figure 6c,e shows the surface distribution maps of Ni and Zr elements, respectively. It can be found that Ni and Zr are still distributed in the pristine layer, without extensive diffusion or fusion.

To clarify the reason for the transition layer, Figure 7a,c analyzes the HRTEM image of the amorphous/transition interface and the crystalline/transition interface, respectively. It is worth mentioning that the film area shown in Figure 7a was irradiated by the electron beam of the transmission electron microscope for 1.5 h when the HRTEM image was taken. Figure 7e shows the bright-field image of the ZrC layer that was taken at this time. The selected area FFT images show a nanocrystals pattern, which is enough to indicate that the amorphous layer was nano-crystallized due to electron irradiation. The HRTEM image reflects that both the Ni crystalline layer and the ZrC amorphous layer are arranged in nanocrystals with different morphologies. However, the transition layer is mainly amorphous where atomic mixing occurs, as shown in the EDS mapping of the irradiated film in Figure 6c,e. The inverse FFT spectrum can confirm that the transition layer is more disordered than the other two layers in Figure 7b,d. Research studies indicate that ion irradiation can boost diffusion and the mixing of elements [51,52,53]. As discussed earlier, the interface of the unirradiated film can be regarded as a mixture of Ni crystals and ZrC amorphous structure, and Ni and Zr elements exist on the same thickness plane near the interface. The irradiation-induced mixing occurs inside irradiation thermal spikes, and the thickness of the amorphous mixed layer is proportional to the square root of the irradiation fluence [33]. Therefore, high-fluence He^+^ irradiation promotes the migration and mixing of atoms, thus aggravating the disorder of the interface region. 

The approximately circular structures with different sizes indicated by the red and yellow arrows in Figure 6b are also noticed. The structure was magnified in TEM bright-field mode at different magnifications and analyzed under both the over-focus and under-focus conditions in Figure 8. The structures were initially determined to be helium bubbles. By comparing the helium bubbles in Figure 8a,b,d,e, it can be found that the number of helium bubbles in the Ni layer is large, and the size ranges from 2 to 15 nm. The reason for the large size difference is that the helium with a high concentration migrates to the nanocrystalline boundary to form larger-size helium bubbles [54]. Meanwhile, the helium bubbles in the amorphous layer are less numerous and smaller in size. Studies have shown that the interface of the crystal film acts as a defect sink, resulting in a much larger helium bubble at the interface [55,56]. It can be seen that the existence of the amorphous layer effectively inhibits the growth of helium bubbles. Meanwhile, almost no helium bubbles were found in the newly formed transition layer, except for both interfaces from the ZrC and Ni sides. The formation of defects around the interface is due to the high diffusion rate of point defects toward the interface in order to decrease the mismatch between layers [54,57]. There are differences in the elastic energy stored in the three layers in the irradiated samples. First, the Ni crystalline layer accommodates elastic energy, which increases at the beginning of the irradiation (due to the formation of the point defects vacancy and interstitials); then, in order to minimize the total energy, the system relaxes by the increase of the bubble size (either via Ostwald or coalescence) [58]. In the amorphous matrix, He-induced defects such as vacancies provide abundant gathering sites for helium introduced by irradiation [59], but the system has already lost its elastic energy; thus, there is an absence of the driving force of bubble growth [60]. Moreover, helium atoms are dispersed throughout the amorphous region, resulting in the high solubility of helium in the amorphous region [61]. Therefore, they do not easily aggregate. However, there are relatively few nanocrystalline interfaces that can serve as trapping sites for helium, and helium atoms are more likely to aggregate into bubbles at limited sites as the irradiation time increases.

## 4. Conclusions

Amorphous-ZrC/nanocrystalline-Ni multilayers with monolayer thicknesses of 5, 10, 20, 50, and 100 nm were prepared by using magnetron sputtering. The films of different monolayer thicknesses still maintained the structure stability even at 50 keV He^+^ irradiation to a fluence of 1.0 × 10^17^ ions/cm^2^ at room temperature. The film with a layer thickness of 100 nm formed an amorphous transition layer of ~30 nm between the amorphous-ZrC and nanocrystalline-Ni layers, due to the irradiation-induced element diffusion or mixing effect. The helium bubbles formed during irradiation are more likely to accumulate and grow in the crystalline layer, while the helium bubbles in the transition layer and the amorphous layer are difficult to grow to be visible. It was preliminarily confirmed that the introduction of an amorphous structure into the thin film material helps to absorb the defects introduced by irradiation, and this can improve the resistance of the material to helium ion irradiation.

## Figures and Tables

**Figure 1 materials-15-03059-f001:**
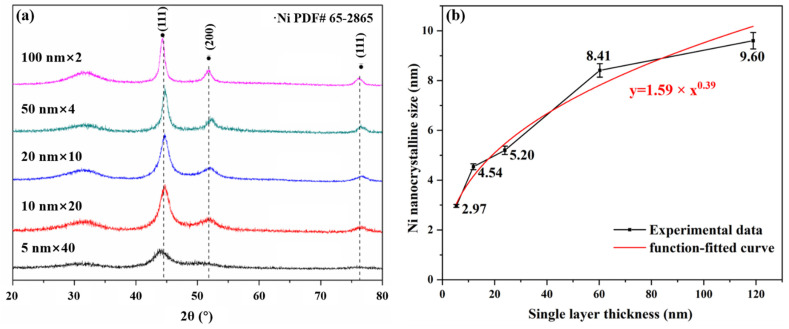
(**a**) The GIXRD spectra of pristine amorphous ZrC/nanocrystalline Ni multilayers. (**b**) The Ni crystalline grain size and its nonlinear fitting curve changing with layer thickness.

**Figure 2 materials-15-03059-f002:**
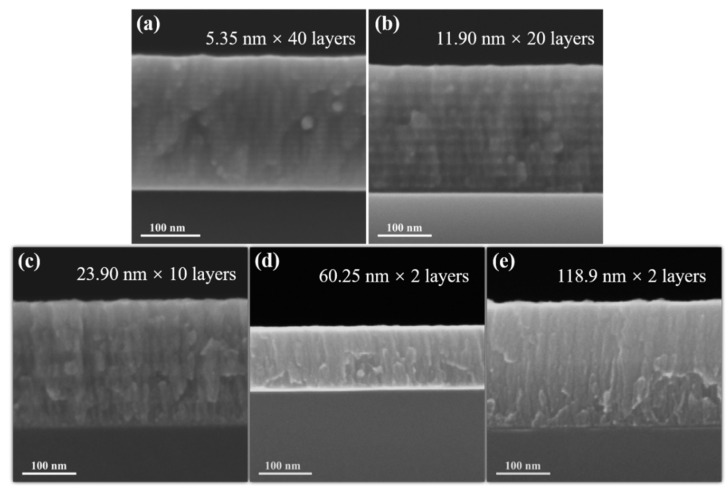
The cross-sectional SEM images of pristine amorphous ZrC/nanocrystalline Ni multilayers: (**a**) 5 nm, (**b**) 10 nm, (**c**) 20 nm, (**d**) 50 nm, and (**e**) 100 nm.

**Figure 3 materials-15-03059-f003:**
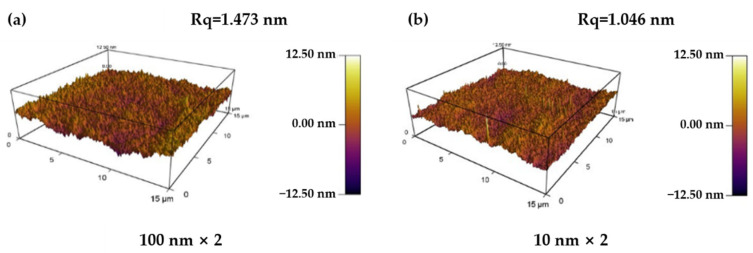
The surface morphology of pristine amorphous ZrC/nanocrystalline Ni multilayers: (**a**) 10 nm and (**b**) 100 nm.

**Figure 4 materials-15-03059-f004:**
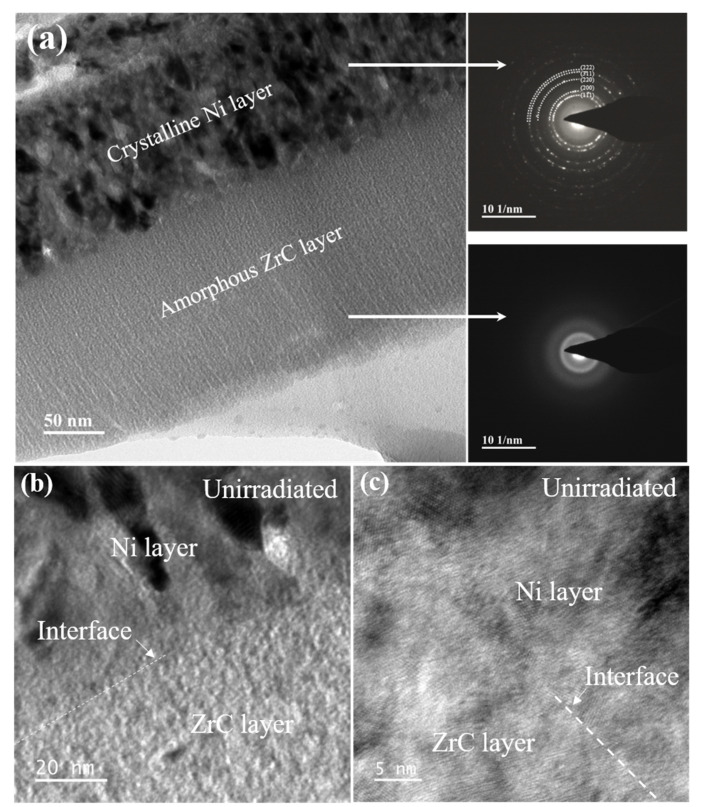
(**a**) The bright-field TEM images and corresponding NBED patterns of pristine amorphous ZrC/nanocrystalline Ni multilayer with single-layer thickness of 100 nm. (**b**) High-magnification bright-field image and (**c**) HRTEM image at the interface of amorphous and crystalline layer.

**Figure 5 materials-15-03059-f005:**
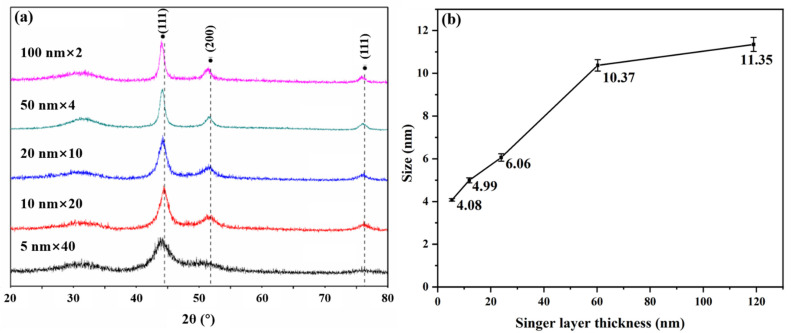
(**a**) The GIXRD spectra of irradiated amorphous ZrC/nanocrystalline Ni multilayer. (**b**) The grain size changing with layer thickness after irradiation.

**Figure 6 materials-15-03059-f006:**
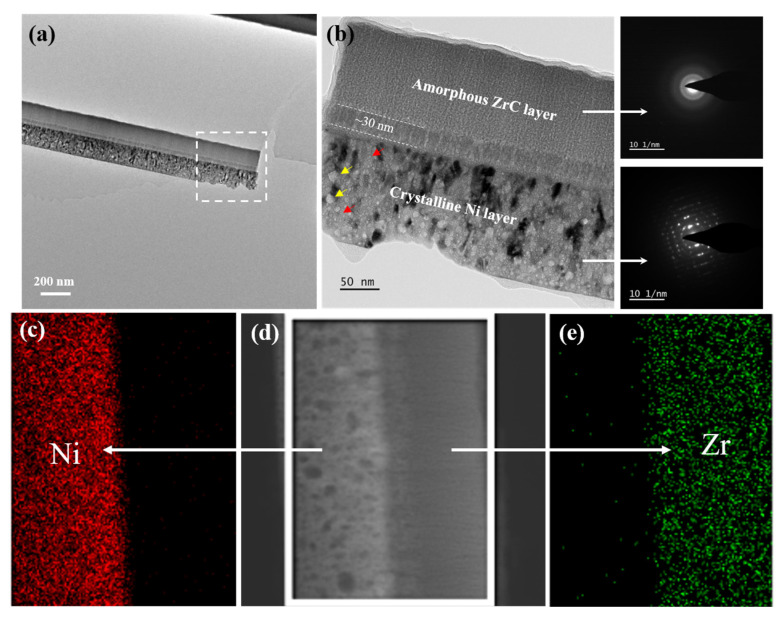
(**a**) Overall topography of the cross-section TEM and (**b**) the bright-field TEM image with corresponding NBED patterns of irradiated amorphous ZrC/nanocrystalline Ni multilayers. (**c**,**e**) EDS mapping of Zr and Ni after irradiation corresponding to (**d**) STEM image.

**Figure 7 materials-15-03059-f007:**
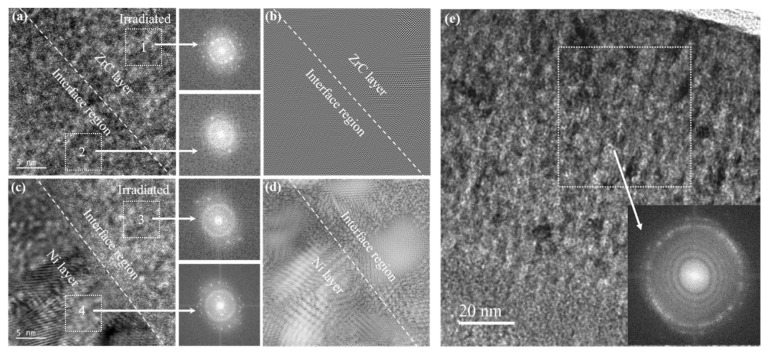
(**a,c**) The HRTEM image with corresponding FFT patterns of the interface on the side of amorphous ZrC and crystalline Ni. (**b,d**) HRTEM image with corresponding FFT patterns of the interface on the side of amorphous ZrC and crystalline Ni. Regions 1, 2, 3, and 4 in the white dashed frames are operated by the FFT transformation. (**e**) The bright-field TEM image with corresponding FFT patterns of ZrC layer after TEM electron-beam focusing for 1.5 h.

**Figure 8 materials-15-03059-f008:**
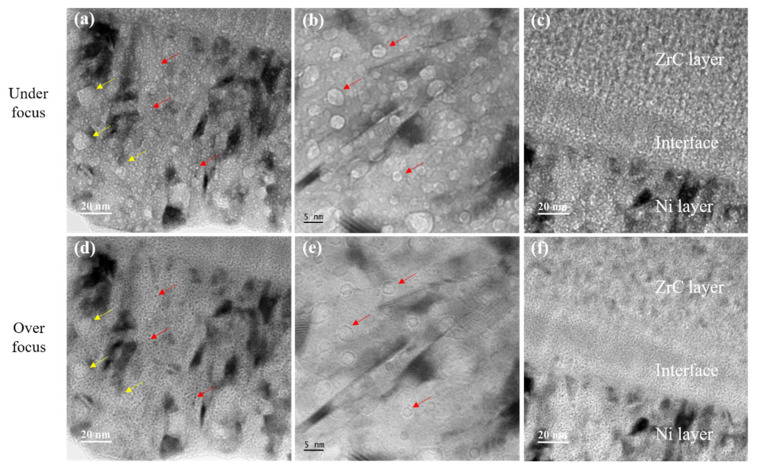
(**a**,**d**) The TEM bright-field images and (**b**,**e**) high-magnification TEM images of the crystalline Ni layer after irradiation. (**c**,**f**) The distribution of helium bubbles in the three regions under the TEM bright-field images. The red and yellow arrows indicate the helium bubbles with different size. (**a**–**c**) Under-focus and (**d**–**f**) over-focus.

**Table 1 materials-15-03059-t001:** Sputtering parameters of amorphous ZrC/nanocrystalline Ni multilayers.

Composition	T_d_ (nm)	Number of Layers	Substrate Temperature	Air Pressure (Pa)	Sputtering Power (W)		Sputtering Rate (Å/s)	
					Ni	ZrC	Ni	ZrC
ZrC/Ni	100	2	Room temperature	~0.7 Pa	80	100	0.91	0.18
	50	4						
	20	10						
	10	20						
	5	40						

**Table 2 materials-15-03059-t002:** Real average layer thickness (T_r_) and grain size in different orientations (D_(hkl)_, D = D_(111)_) of amorphous ZrC/nanocrystalline Ni multilayers.

T_d_/nm	T_r_/nm	D_(111)/_nm	D_(200)/_nm	D_(220)/_nm
5	5.35	2.97	0.68	2.75
10	11.90	4.53	1.80	4.24
20	23.90	5.19	2.66	4.58
50	60.25	8.41	4.67	6.91
100	118.9	9.60	5.86	7.28

## Data Availability

The data is available on reasonable request from the corresponding author.

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
