# Peer review of "The Preparation of Amorphous ZrC/Nanocrystalline Ni Multilayers and the Resistance to He+ Irradiation"

_materials, 2022, doi:10.3390/ma15093059_

Round 1

Reviewer 1 Report

There are a number of minor comments.

  1. Page 2. “In this paper, the magnetron sputtering method was used to deposit ZrC ceramic targets and metal Ni targets alternately on the surface of Si substrates in an Ar atmosphere of 0.7 Pa at room temperature.” Probably meant films, not targets. Please check English one more time.
  2. Please specify the type of power supply for the magnetrons in Section 2.1. RF, DC or pulsed? Also, please specify the size of vacuum chamber, the number of magnetrons, their relative position in the chamber and the size of the sputtered targets.
  3. Why were helium ions chosen to irradiate the samples? On what basis were the irradiation parameters chosen (ion energy and dose)?
  4. Page 3. “Figure 1 shows the GIXRD spectra of pristine amorphous ZrC/nanocrystalline Ni multilayers with different single layer thicknesses and pure Ni film, and the Ni grains size in different films was calculated.” I did not find a diffractogram of the pure nickel film in Fig. 1.
  5. Decipher the abbreviation FWHM at the first time it is mentioned on page 3.
  6. What is the penetration depth of helium ions into the material at 50 keV energy and how does it correlate with film thickness?
  7. The authors found the formation of helium bubbles in the nickel layer after irradiation with helium ions. Has this phenomenon been observed in other works? If possible, compare the results on the radiation resistance of these films with other works.

Author Response

Reviewer 1

  1. Page 2. “In this paper, the magnetron sputtering method was used to deposit ZrC ceramic targets and metal Ni targets alternately on the surface of Si substrates in an Ar atmosphere of 0.7 Pa at room temperature.” Probably meant films, not targets. Please check English one more time.

The corresponding content has been modified and highlighted on page 2 of the manuscript.

  1. Please specify the type of power supply for the magnetrons in Section 2.1. RF, DC or pulsed? Also, please specify the size of vacuum chamber, the number of magnetrons, their relative position in the chamber and the size of the sputtered targets.

The corresponding content has been modified and highlighted on page 2 of the manuscript.

The vacuum chamber is a cylinder with a diameter of 305 mm and a height of 254 mm, and two 2.0” diameter magnetrons confocally configured over a rotating substrate stage and RF or DC power supplies. The silicon wafer and the two targets are distributed in an inverted triangle in the height direction. The silicon wafer is at the bottom and the target is diagonally above it. The distance from silicon wafer to both targets is 80 mm. The diameter of the sputtered targets is 50 mm. Other structural information of magnetron sputtering instruments can be found on the official website.

  1. Why were helium ions chosen to irradiate the samples? On what basis were the irradiation parameters chosen (ion energy and dose)?

The corresponding content has been modified and highlighted on page 3 of the manuscript.

  1. Page 3. “Figure 1 shows the GIXRD spectra of pristine amorphous ZrC/nanocrystalline Ni multilayers with different single layer thicknesses and pure Ni film, and the Ni grains size in different films was calculated.” I did not find a diffractogram of the pure nickel film in Fig. 1.

The corresponding content has been modified and highlighted on page 3 of the manuscript.

This is our mistake. The irradiation experiment of pure nickel has not been done and will not be discussed in this paper.

  1. Decipher the abbreviation FWHM at the first time it is mentioned on page 3.

The corresponding content has been modified and highlighted on page 3 of the manuscript.

  1. What is the penetration depth of helium ions into the material at 50 keV energy and how does it correlate with film thickness?

The corresponding content has been modified and highlighted on page 3 of the manuscript.

  1. The authors found the formation of helium bubbles in the nickel layer after irradiation with helium ions. Has this phenomenon been observed in other works? If possible, compare the results on the radiation resistance of these films with other works.

The corresponding content has been modified and highlighted on page 10 of the manuscript.

Reviewer 2 Report

Dear Authors 

Please see attached 

Good luck

Author Response

Reviewer 2

The results are interesting and important for the investigators and technologists that work under the design and production of the ATF. The authors used different experimental techniques to study the structural properties of ZrC/Ni multilayer with different individual layer thicknesses and the impact of He irradiation at high fluence.

In my opinion, this work fits well in the scope of Materials. Quality of presentation, discussion, and implications meet, overall, with the standards of this Journal. However, there are several issues that the authors should address for the manuscript to be in a suitable form for publication. I provide a detailed list below.

Comments and suggestions

major

  • The introduction should be improved by one paragraph showing the importance of the multilayer to improve the mechanical properties and reduce irradiation damage [Acta Mater. 229 (2022), p. 117807; ACS Appl. Mater. Interfaces 14 (2022), p.12777– 12796.]

The corresponding content has been modified and highlighted on page 1-2 of the manuscript.

2) The introduction should be enriched with strong references such as: Coatings 2020, 10, 835; doi:10.3390/coatings10090835.

Materials 2021, 14(18), 5393; https://doi.org/10.3390/ma14185393

Materials 2019, 12, 1036.

Materials 2020, 13, 794.

Materials 2019, 12, 3343.

Materials 2019, 12, 2639.

Materials 2019, 12, 2628.

Materials 2018, 12, 93.

The corresponding content has been modified and highlighted on page 1-2 and 10-11 of the manuscript.

3) After irradiation, the He-induced damage should induce strain out-of-plane (manifesting in the peak shifts with respect to the virgin one), but the GIXRD results do not show any change which should occur, could the authors measure with a higher incidence angle for example 2° or 5°. If there are peak shifts, the authors should discuss the strain out-of-plane evolution with respect to these papers [Acta Mater. 229 (2022), p. 117807; ACS Appl. Mater. Interfaces 14 (2022), p.12777–12796.].

Since the total thickness of the film is only 200 nm, a higher grazing incidence angle will detect the Bragg peak of the substrate Si, which will mask the diffraction signal of the film. Therefore, the 0.5 degree of grazing angle was used to obtain the X-ray dirraction just from the film compounds, where the intensities of X-ray diffraction is too low.

4) The authors should add the SRIM profile of He distribution and superposed to Fig.6b. Also, the authors should add in the text the Ed displacement energy of Zr, C, and Ni used for SRIM simulation. You can use the displacement energy used in this reference [Acta Mater. 202, (2021) 317–330, Acta Mater. 188 (2020) 609–622].

The corresponding content has been modified and highlighted on page 3 of the manuscript.

5) The discussion about bubbles formation and bubble density, and size in the three regions should be improved and the explanation is not true. I recommend changing the text accordingly:

Meanwhile, almost no helium bubbles were found in the newly formed transition layer except for both interfaces from ZrC and Ni sides. The formation of defects around the interface is due to the high diffusion rate of point defects towards the interface in order to decrease the mismatch between layers. [ACS Appl. Mater. Interfaces 14 (2022), p.12777–12796.].There are differences in the elastic energy stored in the three layers in the irradiated samples. First, the Ni crystalline layer accommodates elastic energy which increases at the beginning of the irradiation (due to the formation of the point defects vacancy and interstitials) then in order to minimize the total energy, the system relax manifesting in the increase of the bubble size (either via Ostwald or coalescence) [1] [ Acta Materialia 181 (2019) 160-172.]. In the amorphous matrix, He-induced defects such as vacancies provide abundant gathering sites for helium introduced by irradiation [34], but the system is already lost its elastic energy, thus the absence of the driving force of bubble growth [Acta Mater. 188 (2020) 609–622.]. Moreover, helium atoms are dispersed throughout the amorphous region, which results in the high solubility of helium in the amorphous region [35]. Therefore, they do not easily aggregate. However, there are relatively few nanocrystalline interfaces that can serve as trapping sites for helium, and helium atoms are more likely to aggregate into bubbles at limited sites as the irradiation time increases.

The reviewer's comments can better reveal the underlying mechanism of helium distribution in the three layers, and the article has been revised. The corresponding content has been modified and highlighted on 10 of the manuscript.

Minor

Fig.1b should be corrected Y-axis (single layer thickness), the red line is fitting and the black line is exp.

The corresponding content has been modified on page 4 of the manuscript.

Fig.3, could the authors explain why the surface height varies from -12.5 nm to 12.5 nm? Change the title of the manuscript

“-12.5 nm to 12.5 nm” only represents the surface height values corresponding to the colors of different depths, and does not represent the surface height of a certain material.

Fig.4 the authors talk about the layer thickness of 100 nm but from the TEM images the layer thickness is around 200 nm, please check it, or maybe is a mistake in the scale

“100 nm” is the single layer thickness of the film, and the total thickness of the film is 200 nm. The corresponding content has been modified on page 6 of the manuscript.

Reviewer 3 Report

The authors have provided a paper with important data that concerns an significant problem in the nuclear industry.  The technical content as it is presented is fairly thorough and appears to be sound.  There are several key points that should be addressed before the paper is published:

  1. The authors switch between using “dose” and “fluence” when describing the magnitude of the He irradiation (1.0x1017 ions/cm2). Given the distinct meaning of dose in many areas of radiation effects as it relates to the energy deposition in a material, it is better and correct to use the term fluence. The authors should use fluence consistently throughout the paper.
  2. The authors should give reasons why this fluence was chosen. Does it have operational significance for actual nuclear fuel applications? If so, what does it represent? Did this fluence have something to do with the Fukushima incident?  If not these things then why this fluence? Was it simply the highest practical fluence that could be used?
  3. The authors need to describe how the sputter deposition rate was measured and how the desired film thickness was obtained in the Sample Preparation section. As later noted in Table 2, the measured thicknesses were consistently higher than the desired thickness. Did this consistent difference have to do with the sputtering process?
  4. The authors choose to concentrate analysis on the nominally 100 nm film, but it is never completely clear why this is so. The thinner films have been made and appear to have been irradiated. However, the thinner films are not discussed to any extent in the paper.  The authors need to make clear why their focus was on the 100 nm film.  Was this only because of the radiation induced interface region in the 100 nm sample?  If so, this needs to be explicitly discussed.  In addition, the authors need to at least briefly discuss the results from the thinner films, even if it is only to justify why they focus on the 100 nm film.
  5. The caption of Figure 7 needs work. Fig 7 (d) is not discussed clearly and it is not clear what is meant by "region 1, 2, 3 and 4". The authors need to make clearer the purpose of this data.
  6. The paragraph directly following Figure 7 is not well written. Please see:

“with size ranging from 2 nm to 15 nm compared to Figure 8a,b,d,e.” To what is the word “compared” referring to in this sentence.  The meaning of this sentence needs to be made clearer.

“While the helium bubbles in the amorphous layer are less numerous and smaller in size, comparable to the helium bubble shown by the red arrows in the crystalline layer.”  This sentence is confusing. Are the authors saying the bubbles are different or comparable? Comparable how?

“Just as indicated by the red arrows in ZrC layer from Figure 8c,f.”  This is a sentence fragment and its meaning is not clear.  The sentence seems to indicate that there should be arrows in Figure 8 c,f, but there are no arrows.

  1. In the conclusion, the authors need to relate their results back to their original motivation. Are they indeed concluding that the layered materials will provide enhance protection with regard to the Zr/ high temperature water issue? If so this needs to be explicitly stated and explained.

Author Response

Reviewer 3:

The authors have provided a paper with important data that concerns an significant problem in the nuclear industry.  The technical content as it is presented is fairly thorough and appears to be sound.  There are several key points that should be addressed before the paper is published:

  1. The authors switch between using “dose” and “fluence” when describing the magnitude of the He irradiation (1.0x1017 ions/cm2). Given the distinct meaning of dose in many areas of radiation effects as it relates to the energy deposition in a material, it is better and correct to use the term fluence. The authors should use fluence consistently throughout the paper.

The dose have been changed to fluence through the whole manuscript.

  1. The authors should give reasons why this fluence was chosen. Does it have operational significance for actual nuclear fuel applications? If so, what does it represent? Did this fluence have something to do with the Fukushima incident?  If not these things then why this fluence? Was it simply the highest practical fluence that could be used?

The corresponding content has been modified on page 3 of the manuscript.

  1. The authors need to describe how the sputter deposition rate was measured and how the desired film thickness was obtained in the Sample Preparation section. As later noted in Table 2, the measured thicknesses were consistently higher than the desired thickness. Did this consistent difference have to do with the sputtering process?

First, pre-sputtering was performed, and ZrC films with a certain thickness were deposited for a fixed time, and the film thickness was measured by SEM. From this, the thickness of sputtering per unit time is obtained, that is, the sputtering rate, and the same operation is performed for the Ni thin film. The desired thickness is obtained by controlling the time during the actual sputtering process. The actual thickness is affected by slight changes of air pressure during sputtering, or human factors (manual timing, target switching, etc.), and there is no regular difference in theory. The data discrepancy consistency in this paper may just be a coincidence.

  1. The authors choose to concentrate analysis on the nominally 100 nm film, but it is never completely clear why this is so. The thinner films have been made and appear to have been irradiated. However, the thinner films are not discussed to any extent in the paper.  The authors need to make clear why their focus was on the 100 nm film.  Was this only because of the radiation induced interface region in the 100 nm sample?  If so, this needs to be explicitly discussed.  In addition, the authors need to at least briefly discuss the results from the thinner films, even if it is only to justify why they focus on the 100 nm film.

The article proves that all materials have no phase transition during GIXRD characterization, and the phenomenon is basically the same. Therefore, films with single layer thickness of 100 nm were chosen to characterize the microstructure in order to see other interesting phenomena in films of larger size.

  1. The caption of Figure 7 needs work. Fig 7 (d) is not discussed clearly and it is not clear what is meant by "region 1, 2, 3 and 4". The authors need to make clearer the purpose of this data.

The picture has been modified on page 10 of the manuscript.

  1. The paragraph directly following Figure 7 is not well written. Please see:

“with size ranging from 2 nm to 15 nm compared to Figure 8a,b,d,e.” To what is the word “compared” referring to in this sentence.  The meaning of this sentence needs to be made clearer.

The corresponding content has been modified on page 10 of the manuscript. Here, the over-focus (a,b) and under-focus (d,e) photos are compared to estimate the size of the helium bubbles in the nickel layer.

“While the helium bubbles in the amorphous layer are less numerous and smaller in size, comparable to the helium bubble shown by the red arrows in the crystalline layer.”  This sentence is confusing. Are the authors saying the bubbles are different or comparable? Comparable how?

The corresponding content has been modified on page 10 of the manuscript. It is emphasized that the number of visible helium bubbles in the amorphous layer is small and the size is only similar to the smallest size helium bubbles in the crystalline layer.

“Just as indicated by the red arrows in ZrC layer from Figure 8c,f.”  This is a sentence fragment and its meaning is not clear.  The sentence seems to indicate that there should be arrows in Figure 8 c,f, but there are no arrows.

This sentence is redundant and has been deleted.

  1. In the conclusion, the authors need to relate their results back to their original motivation. Are they indeed concluding that the layered materials will provide enhance protection with regard to the Zr/ high temperature water issue? If so this needs to be explicitly stated and explained.

The corresponding content has been modified on page 11 of the manuscript.

Round 2

Reviewer 2 Report

Dear editor 

The authors react with most of my concerns, I recommend to accept the paper as it is

BR